# Authentic self-expression on social media is associated with greater subjective well-being

Erica R. Bailey [1,3✉], Sandra C. Matz [1,3], Wu Youyou[2] & Sheena S. Iyengar[1]

Social media users face a tension between presenting themselves in an idealized or authentic way. Here, we explore how prioritizing one over the other impacts users' well-being. We estimate the degree of self-idealized vs. authentic self-expression as the proximity between a user's self-reported personality and the automated personality judgements made on the basis Facebook Likes and status updates. Analyzing data of 10,560 Facebook users, we find that individuals who are more authentic in their self-expression also report greater Life Satisfaction. This effect appears consistent across different personality profiles, countering the proposition that individuals with socially desirable personalities benefit from authentic self-expression more than others. We extend this finding in a pre-registered, longitudinal experiment, demonstrating the causal relationship between authentic posting and positive affect and mood on a within-person level. Our findings suggest that the extent to which social media use is related to well-being depends on how individuals use it.

[1] Columbia Business School, Columbia University, 3022 Broadway, New York, NY 10027, USA. [2] Kellogg School of Management, Northwestern University, 2211 Campus Dr, Evanston, IL 60208, USA. [3]These authors contributed equally: Erica R. Bailey, Sandra C. Matz. ✉email: Erica.bailey@columbia.edu

Social media can seem like an artificial world in which people's lives consist entirely of exotic vacations, thriving friendships, and photogenic, healthy meals. In fact, there is an entire industry built around people's desire to present idealistic self-representations on social media. Popular applications like FaceTune, for example, allow users to modify everything about themselves, from skin tone to the size of their physical features. In line with this "self-idealization perspective", research has shown that self-expressions on social media platforms are often idealized, exaggerated, and unrealistic[1]. That is, social media users often act as virtual curators of their online selves[2] by staging or editing content they present to others[3].

A contrasting body of research suggests that social media platforms constitute extensions of offline identities, with users presenting relatively authentic versions of themselves[4]. While users might engage in some degree of self-idealization, the social nature of the platforms is thought to provide a degree of accountability that prevents individuals from starkly mis-representing their identities[5]. This is particularly true for plat-forms such as Facebook, where the majority of friends in a user's network also have an offline connection[6]. In fact, modern social media sites like Facebook and Instagram are far more realistic than early social media websites such as Second Life, where users presented themselves as avatars that were often fully divorced from reality[7]. In line with this authentic self-expression per-spective, research has shown that individuals on Facebook are more likely to express their actual rather than their idealized personalities[8,9].

The desire to present the self in a way that is ideal and authentic is not mutually exclusive; on the contrary, an individual is likely to desire both simultaneously[10]. This occurs in part because self-idealization and authentic self-expression fulfill dif-ferent psychological needs and are associated with different psychological costs. On the one hand, self-idealization has been called a "fundamental part of human nature"[11] because it allows individuals to cultivate a positive self-view and to create positive impressions of themselves in others[12]. In addition, authentic self-expression allows individuals to verify and affirm their sense of self[13,14] which can increase self-esteem[15], and a sense of belonging[16]. On the other hand, self-idealizing behavior can be psychologically costly, as acting out of character is associated with feelings of internal conflict, psychological discomfort, and strong emotional reactions[17,18]; individuals may also possess character-istics that are more or less socially desirable, bringing their desire to present themselves in an authentic way into conflict with their desire to present the best version of themselves.

Here, we explore the tension between self-idealization and authentic self-expression on social media, and test how prior-itizing one over the other impacts users' well-being. We focus our analysis on a core component of the self: personality[19]. Per-sonality captures fundamental differences in the way that people think, feel and behave, reflecting the psychological characteristics that make individuals uniquely themselves[20,21]. Building on the Five Factor Model of personality[22], we test the extent to which authentic self-expression of personality characteristics are related to Life Satisfaction, hypothesizing that greater authentic self-expression will be positively correlated with Life Satisfaction. In exploratory analyses, we also consider whether this relationship is moderated by the personality characteristics of the individual. That is, not all individuals might benefit from authentic self-expression equally. Given that some personality traits are more socially desirable than others[23], individuals who possess more desirable personality traits are likely to experience a reduced tension between self-idealization and authentic self-expression. Consequently, individuals with more socially desirable profiles might disproportionality benefit from authentic self-expression

because the motivational pulls of self-idealization and authentic self-expression point in the same—rather than the opposite—direction.

Previous literature on authentic self-expression has pre-dominantly relied on self-reported perceptions of authenticity as (i) a state of feeling authentic[24], or (ii) a judgement about the honesty or consistency of one's self[25]. However, such self-reported measures have been shown to be biased by valence states, and social desirability[26,27]. To overcome these limitations, in Study 1 we introduce a measure of Quantified Authenticity. If authenticity is most simply defined as the unobstructed expres-sion of one's self[28], then authenticity can be estimated as the proximity of an individual's self-view and their observable self-expression. We calculate Quantified Authenticity by comparing self-reported personality to personality judgements made by computers on the basis of observable behaviors on Facebook (i.e., Likes and status updates).

By observing self-presentation on social media and comparing it to the individual's self-view, we are able to quantify the extent to which an individual deviates from their authentic self. That is, we locate each individual on a continuum that ranges from low authenticity (i.e., large discrepancy between the self-view and observable self-expression) to high authenticity (i.e., perfect alignment between the self-view and observable self-expression). Importantly, our approach rests on the assumption that any deviation from the self-view on social media constitutes an attempt to present oneself in a more positive light, and therefore a form of self-idealization. While a deviation could theoretically indicate both self-idealization and self-deprecation, it is unlikely that users will deviate from their true selves in a way that makes them look worse in the eyes of others. A strength of our measures is that we do not postulate that self-idealization takes a particular form of deviation from the self or is associated with striving for a particular profile. Although research suggests that there are cer-tain personality traits that are more desirable on average[29,30], the extent to which a person sees scoring high or low on a given trait is likely somewhat idiosyncratic and depends—at least in part—on other people in their social network. For example, behaving in a more extraverted way might be self-enhancing for most people; however, there might be individuals for whom behaving in a more introverted way might be more desirable (e.g. because the norm of their social network is more introverted). Hence, our conceptualization of Quantified Authenticity allows for deviations in different directions (see Supplementary Information for more detail).

## Results

**Quantified Authenticity and subjective well-being.** In Study 1, we analyzed the data of 10,560 Facebook users who had com-pleted a personality assessment and reported on their Life Satis-faction through the myPersonality application[31,32]. To estimate the extent to which their Facebook profiles represent authentic expressions of their personality, we compared their self-ratings to two observational sources: predictions of personality from Face-book Likes ($N = 9237$)[33] and predictions of personality from Facebook status updates ($N = 3215$)[34]. These are based on recent advances in the automatic assessment of psychological traits from the digital traces they leave on Facebook[35]. For each of the observable sources, we calculated Quantified Authenticity as the inverse Euclidean distance between all five self-rated and obser-vable personality traits. Our measure of Quantified Authenticity exhibits a desirable level of variance, ranging all the way from highly authentic self-expression to considerable levels of self-idealization (see ridgeline plot of Quantified Authenticity calcu-lated for self-language and Self-Likes in Supplementary Fig. 3, see

**Table 1 Regression analysis of Life Satisfaction on Quantified Authenticity.**

| | Model 1 | | | Likes Model<br>Model 2 | | | Model 3 | | |
|---|---|---|---|---|---|---|---|---|---|
| | *B* | SE(*B*) | β | *B* | SE(*B*) | β | *B* | SE(*B*) | β |
| QA | 0.156*** | 0.014 | 0.113 | 0.069*** | 0.013 | 0.050 | 0.070*** | 0.013 | 0.051 |
| Extr. | – | – | – | 0.027* | 0.013 | 0.020 | 0.022 | 0.014 | 0.016 |
| O | – | – | – | −0.025* | 0.013 | −0.018 | −0.016 | 0.013 | −0.012 |
| C | – | – | – | 0.161*** | 0.013 | 0.117 | 0.165*** | 0.014 | 0.120 |
| E | – | – | – | 0.151*** | 0.014 | 0.109 | 0.155*** | 0.014 | 0.112 |
| A | – | – | – | 0.080*** | 0.014 | 0.058 | 0.081*** | 0.014 | 0.059 |
| N | – | – | – | −0.511*** | 0.015 | −0.371 | −0.517*** | 0.015 | −0.375 |
| QA × O | – | – | – | – | – | – | 0.026 | 0.014 | 0.018 |
| QA × C | – | – | – | – | – | – | 0.010 | 0.014 | 0.008 |
| QA × E | – | – | – | – | – | – | 0.014 | 0.014 | 0.011 |
| QA × A | – | – | – | – | – | – | −0.002 | 0.013 | −0.001 |
| QA × N | – | – | – | – | – | – | −0.021 | 0.016 | −0.015 |
| Adj-$R^2$ | | 0.01 | | | 0.26 | | | 0.26 | |
| | | | | **Language Model** | | | | | |
| QA | 0.119*** | 0.025 | 0.085 | 0.046* | 0.023 | 0.032 | 0.059* | 0.024 | 0.042 |
| Extr. | – | – | – | 0.005 | 0.023 | 0.003 | 0.001 | 0.024 | 0.001 |
| O | – | – | – | −0.029 | 0.022 | −0.021 | −0.019 | 0.023 | −0.013 |
| C | – | – | – | 0.157*** | 0.023 | 0.111 | 0.159*** | 0.024 | 0.113 |
| E | – | – | – | 0.150*** | 0.024 | 0.106 | 0.153*** | 0.024 | 0.109 |
| A | – | – | – | 0.123*** | 0.023 | 0.087 | 0.119*** | 0.024 | 0.085 |
| N | – | – | – | −0.503*** | 0.026 | −0.357 | −0.516*** | 0.026 | −0.366 |
| QA × O | – | – | – | – | – | – | 0.027 | 0.021 | 0.022 |
| QA × C | – | – | – | – | – | – | −0.005 | 0.023 | −0.004 |
| QA × E | – | – | – | – | – | – | 0.022 | 0.023 | 0.016 |
| QA × A | – | – | – | – | – | – | −0.015 | 0.024 | −0.011 |
| QA × N | – | – | – | – | – | – | −0.059 | 0.027 | −0.045 |
| Adj-$R^2$ | | 0.01 | | | 0.25 | | | 0.25 | |

Models presented include Quantified Authenticity (QA-Euclidean Distance) on Life Satisfaction, with controls, and interaction effects of QA and the Big Five personality traits. (Likes-Based Model $N = 9237$; Language-Based Model $N = 3215$). ***$p < 0.001$, **$p < 0.01$, *$p < 0.05$.

Supplementary Tables 1 and 2 for zero-order correlations among variables).

To test the extent to which authentic self-expression is related to Life Satisfaction, we ran linear regression analyses predicting Life Satisfaction from the two measures of Quantified Authenticity (Likes, status updates). The results support the hypothesis that higher levels of authenticity (i.e. lower distance scores) are positively correlated with Life Satisfaction (Table 1, Model 1 without controls). These effects remained statistically significant when controlling for self-reported personality traits. Additionally, we included a control variable for the overall extremeness of an individual's personality profile (deviation from the population mean across all five traits), as people with more extreme personality profiles might find it more difficult to blend into society and therefore experience lower levels of well-being[36] (see Table 1, Model 2 with controls; the results are largely robust when controlling for gender and age, see Supplementary Table 3; see Supplementary Figs. 1 and 2 for interactions between individual self-reported and predicted personality traits).

To further explore the mechanisms of Quantified Authenticity, we conducted analyses that distinguished between normative self-enhancement (i.e., rating oneself as more Extraverted, Agreeable, Conscientiousness, Emotionally Stable, and Open-minded than is indicated by one's Facebook behavior) from self-deprecation (i.e., rating oneself lower on all of these traits). While normative self-enhancement has a negative effect on well-being, normative self-deprecation has no effect. These findings suggest that self-enhancement specifically, rather than overall self-discrepancy/

lack of authenticity, is detrimental to subjective well-being (see Supplementary Fig. 4).

To test the robustness of our effects, we regressed Life Satisfaction on three additional measures of Quantified Authenticity (i.e., calculated using Manhattan Distance, Cosine Similarity, and Correlational Similarity; see SI for details on these measures). In both comparison sets (likes and status updates), we found significant and positive correlations between the various ways of estimating Quantified Authenticity (see Supplementary Tables 1 and 2). The standardized beta-coefficients across all four metrics of Quantified Authenticity and observable sources are displayed in Fig. 1. Despite variance in effect sizes across measures and model specifications, the majority of estimates are statistically significant and positive (11 out of 16). Importantly, no coefficients were observed in the opposite direction. These results suggest that those who are more authentic in their self-expression on Facebook (i.e., those who present themselves in a way that is closer to their self-view) also report higher levels of Life Satisfaction.

In exploratory analyses, we considered whether authenticity might benefit individuals of different personalities differentially. In order to examine this, we regressed Life Satisfaction on the interactions between Quantified Authenticity and each of the five personality traits (e.g., Quantified Authenticity × Extraversion). The results of these interaction analyses did not provide reliable evidence for the proposition that individuals with socially desirable profiles (i.e., high openness, conscientiousness, extraversion, agreeableness, and low neuroticism) benefit from

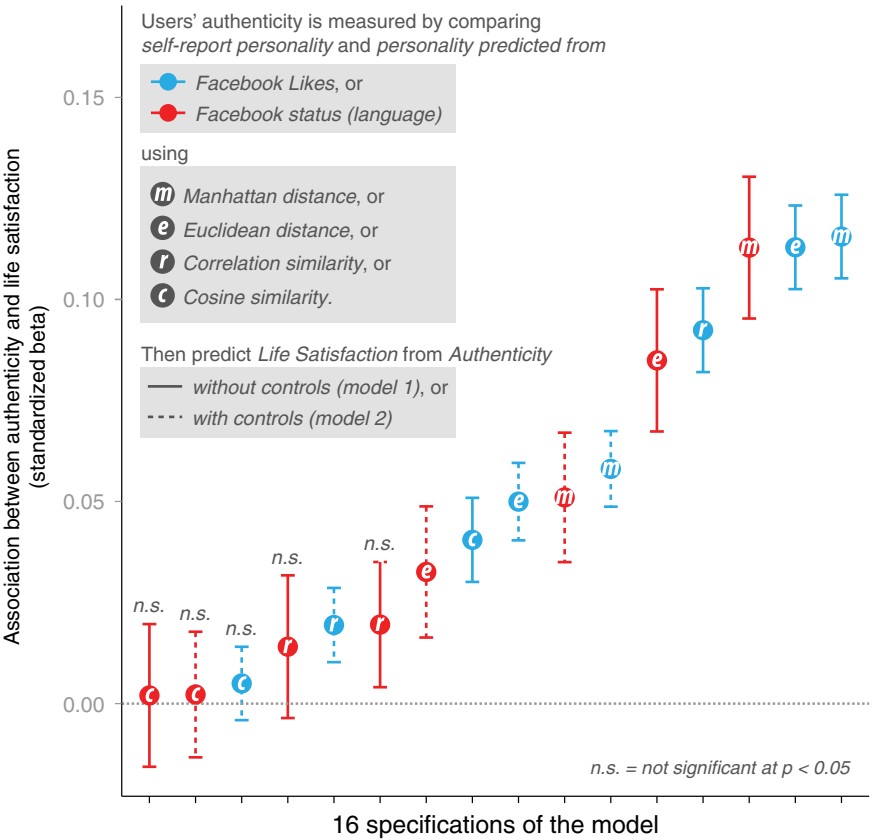

**Fig. 1 Quantified authenticity on Life Satisfaction.** Figure 1 presents standardized beta coefficients for Quantified Authenticity using ordinary least squares regressions in 16 individual regressions predicting Life Satisfaction. Quantified Authenticity is significantly associated with Life Satisfaction in 11 out of the 16 models. Quantified Authenticity is measured as the consistency between self-reported personality and two other sources of personality data: language and Likes, respectively, (indicated in red and blue color). Quantified Authenticity is defined using four distance metrics, respectively: Manhattan, Euclidean, correlation, and cosine similarity (indicated with a letter in the dots). Models with and without control variables are indicated with dashed and solid line, respectively.

authentic self-expression more than individuals with less socially desirable profiles (see Table 1, Model 3). While the interactions of the five personality traits with Quantified Authenticity reached significance for some traits and measures, the results were not consistent across both observable sources of self-expression (Likes-based and Language-based). Consequently, we did not find reliable evidence that having a socially desirable personality profile boosts the effect of authenticity on well-being. Instead, individuals reported increased Life Satisfaction when they presented authentic self-expression, regardless of their personality profile.

The findings of Study 1 provide evidence for the link between authenticity on social media and well-being in a setting of high external validity. However, given the correlational nature of the study, we cannot make any claims about the causality of the effects. While we hypothesize that expressing oneself authentically on social media results in higher levels of well-being, it is also plausible that individuals who experience higher levels of well-being are more likely to express themselves authentically on social media. To provide evidence for the directionality of authenticity on well-being, we conducted a pre-registered, longitudinal experiment in Study 2 (see Fig. 2 for an illustration of the experimental design).

**Experimental manipulation of authentic self-expression on well-being.** We recruited 90 students and social media users at a Northeastern University to participate in a 2-week study

($M_{age} = 22.98$, $SD_{age} = 4.17$, 72.22% female). The sample size deviates from our pre-registered sample size of 200. The reason for this is that the behavioral research lab of the university was shut down after the first wave of data collection due to the COVID-19 pandemic.

All participants completed two intervention stages during which they were asked to post on their social media profiles in a way that was: (1) authentic for 7 days and (2) self-idealized for 7 days. The order in which participants completed the two interventions was randomly assigned. This experimental set-up allowed us to study the effects of authentic versus idealized self-expression on social media in between-person (week 1) and within-person analyses (comparison between week 1 and week 2). All analyses were pre-registered prior to data collection[37]. Given the reduced sample size, the effects reported in this paper are all as expected in effect size, but only partially reached significance at the conventional alpha = 0.05 level. Consequently, we also consider effects that reach significance at alpha = 0.10 as marginally significant.

All participants completed a personality pre-screen (IPIP)[38] prior to beginning the study, and received personalized feedback report at the beginning of the treatment period (t0). Both the authentic and self-idealized interventions (see Methods for details) asked participants to reflect on that feedback report and identify specific ways in which they could alter their self-expression on social media to align their posts more closely with their actual personality profile (authentic intervention) or to align their posts more closely with how they wanted to be seen by

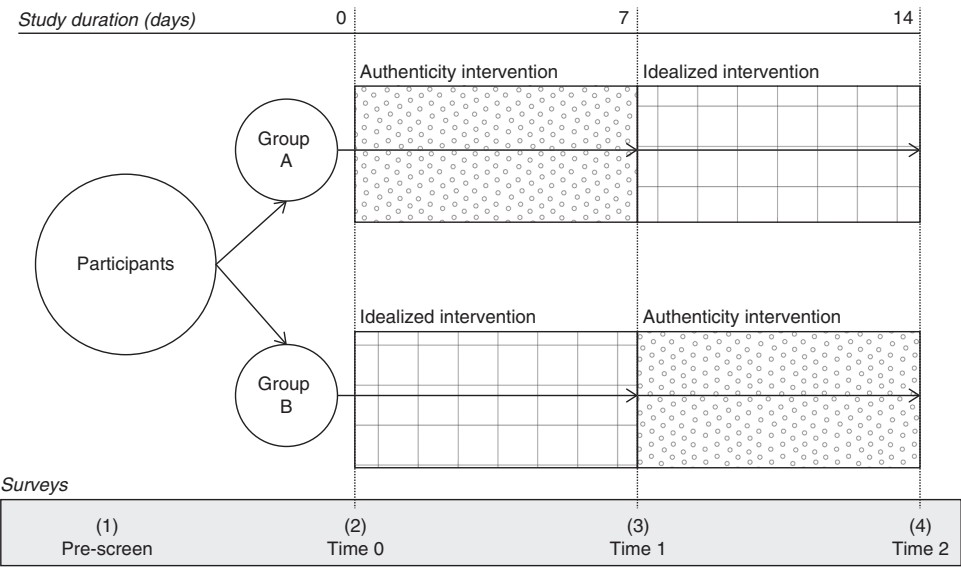

**Fig. 2 Study 2 experimental design.** Figure 2 presents the longitudinal experimental study design for Study 2 with key timepoints, interventions, and surveys.

others (see Supplementary Information for treatment text and examples of responses). The operationalization of the treatment follows our conceptualization of Quantified Authenticity in Study 1 in that it does not prescribe the direction of personality change (e.g. towards higher levels of extraversion). Instead, this design leaves it up to participants what posting in a more desirable way means in relation to their current profile.

Participants self-reported their subjective well-being as Life Satisfaction[39], a single-item mood measure, and positive and negative affect[40] a week after the first intervention (t1), and a week after the second intervention (t2). This design allowed us to examine the causal nature of posting for a week in which participants posted authentically ("authentic, real, or true"), compared to a week in which they posted in a self-idealized way ("ideal, popular or pleasing to others"). Specifically, we hypothesized that individuals who post more authentically over the course of a week would self-report greater subjective well-being at the end of that week, both at the between and within-person level.

We examined the effect of authentic versus self-idealized expression at the between person level at t1 (see t1 in Fig. 3) using independent $t$-tests. Contrary to our expectations, we did not find any significant differences between the two conditions for any of the well-being indicators. This suggests that individuals in the authentic vs. self-idealized conditions did not differ from one another in their level of well-being after the first week of the study. However, when examining the effect within subjects using dependent $t$-tests we found that participants reported significantly higher levels of well-being after the week in which they posted authentically as compared to the week in which they posted in a self-idealized way. Specifically, the well-being scores in the authentic week were found to be significantly higher than in the self-idealized week for mood (mean difference = 0.19 [0.003, 0.374], $t = 2.02$, $d = 0.43$, $p = 0.046$) and for positive affect (mean difference = 0.17 [0.012, 0.318], $t = 2.14$, $d = 0.45$, $p = 0.035$), and marginally significant for negative affect (mean difference = −0.20 [−0.419, 0.016], $t = −1.84$, $d = 0.39$, $p = 0.069$). There was no significant effect on Life Satisfaction (mean difference = 0.09 [−0.096, 0.274], $t = 0.96$, $d = 0.20$, $p = 0.342$).

These findings are reflected in Fig. 3 which showcases the interactions between condition and time point. The graphs

highlight that subjective well-being was higher in the weeks in which participants were asked to post authentically (red bars) compared to those in which they were asked to post in a self-idealized way (blue bars). While there was no difference in subjective well-being across conditions at t1, subjective well-being measures differed significantly between the authentic and self-idealized conditions at t2. We found no significant difference between conditions on Life Satisfaction (mean difference = 0.29 [−0.226, 0.798], $t = 1.11$, $d = 0.23$, $p = 0.270$), however, we found a significant difference between conditions such that the group which received the authenticity treatment had greater positive affect (mean difference = 0.45 [0.083, 0.825], $t = 2.43$, $d = 0.51$, $p = 0.017$), lower negative affect (mean difference = −0.57 [−1.034, −0.113], $t = −2.47$, $d = 0.52$, $p = 0.015$), and higher overall mood (mean difference = 0.40 [0.028, 0.775], $t = 2.14$, $d = 0.45$, $p = 0.036$).

The findings of the experiment provide support for the causal relationship between posting authentically, compared to posting in a self-idealized way, on the more immediate affective indicators of subjective-wellbeing, including mood and affect, but not on the more long-term, cognitive indicator of life satisfaction. This findings aligns with our pre-registration in that we had predicted mood and affect measures to be more sensitive to the treatment compared to Life Satisfaction, which is a broader global assessment one's overall life[39] and less likely to change in the course of a week.

Additionally, the fact that we did not find significant effects in our between-subjects analysis in the first week of the study suggests that authentic self-expression might be difficult to manipulate in a one-off treatment as social media users are likely used to expressing themselves on social media both authentically and in a self-idealized way. Thus, when only one strategy is emphasized, participants might not shift their behavior. This is supported by the finding that participants did not differ significantly in their subjective experience of authenticity on social media at t1 (mean in authentic condition at t1 = 5.56, mean in self-idealized condition at t1 = 5.55, $t = 0.05$, $d = 0.01$, $p = 0.958$; Participants responded to a single item, which read "This past week, I was authentic on social media" on a 7-point scale where 1 = strongly disagree and 7 = strongly agree), indicating that the between-subjects

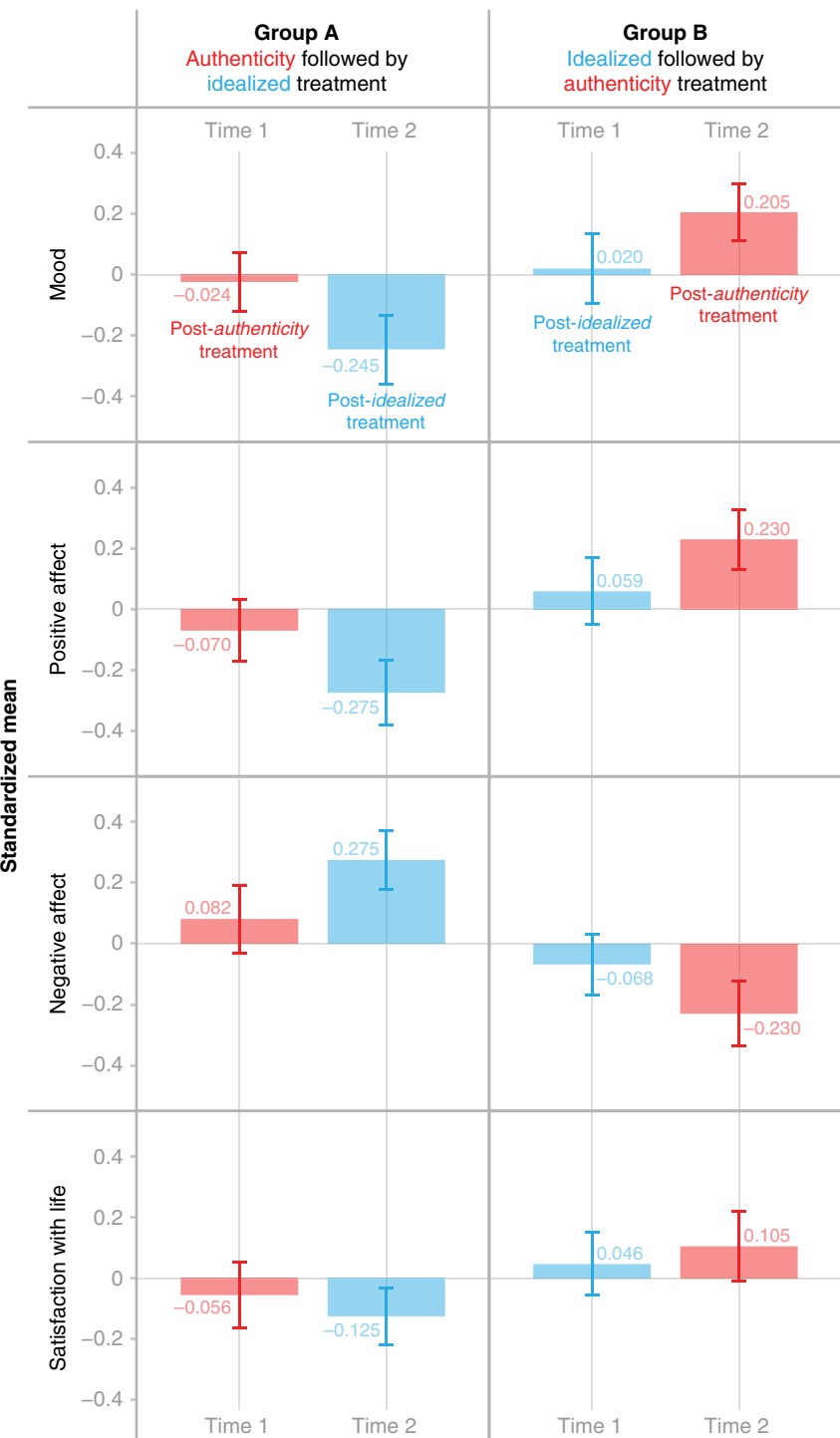

**Fig. 3 Experimental outcomes by authenticity and idealization treatment.** The bar chars illustrate the standardized mean of well-being indicators (mood, positive affect, negative affect, and Life Satisfaction) across two study time points by condition. The red bars indicate scores for the weeks in which participants were asked to post authentically, and the blue bars scores for the weeks in which they were asked to post in a self-idealized way. Error bars represent standard errors. The left-side panel presents Group A who received the authenticity treatment followed by the idealized treatment. The right-side panel presents Group B who received the idealized treatment followed by the authenticity treatment. This experiment was conducted once with independent samples in each group.

manipulation was unsuccessful in getting people to shift their behaviors more toward self-idealized or authentic self-expression compared to their baseline. However, the contrast of the two strategies highlighted in the within-subjects part of the study seems to have successfully shifted participants'

behavior. When compared within person, students did indeed report higher levels of experienced authenticity in their posting during the week in which they were instructed to post authentically (mean difference = 0.30 [0.044, 0.556], $t = 2.33$, $d = 0.49$, $p = 0.022$).

## Discussion

We often hear the advice to just be ourselves. Indeed, psychological theories have suggested that behaving in a way that is consistent with the self-view is beneficial for individual well-being[41]. However, prior investigations of authenticity and well-being have relied solely on self-reported measures which can be confounded by valence and social desirability biases. We estimated authenticity as the proximity between the self-view and self-expression on social media—which we termed Quantified Authenticity—and found that authentic self-expression on social media was correlated with greater Life Satisfaction, an important component of overall well-being. This effect was robust across two comparison points, computer modeled personality based on Facebook Likes and status updates. Our findings suggest that if users engage in self-expression on social media, there may be psychological benefits associated with being authentic. We replicate this finding in a longitudinal experiment with university students; being prompted to post in an authentic way was associated with more positive mood and affect, and less negative mood within participants. Contrary to our second hypothesis, we did not find consistent support for interactions between personality traits and authenticity, such that individuals with more socially desirable traits would benefit more from behaving authentically. Instead, our findings suggest that all individuals regardless of personality traits could benefit from being authentic on social media.

Our findings contribute to the existing literature by speaking directly to conflicting findings on the effects of social media use on well-being. Some studies find that social media use increases self-esteem and positive self-view[42], while others find that social media use is linked to lower well-being[43]. Still, others find that the effect of social media on well-being is small[44] or non-existent[45]. In an attempt to reconcile these mixed findings, researchers have suggested that the extent to which social media platforms related to lower or higher levels of well-being might depend not on whether people use them but on how they use them. For example, research has shown that active versus passive Facebook use has divergent effects on well-being. While passively using Facebook to consume the content share by others was negatively related to well-being, actively using Facebook to share content and communicate was not[46]. We add to this growing body of research by suggesting that effects of social media use on well-being may also be explained by individual differences in self-expression on social media.

Our study has a number of limitations that should be addressed by future research. First, our analyses focused exclusively on the effects of authentic social media use on well-being, and cannot speak to the question of whether an authentic social media use is better or worse than not using social media at all. That is, even though using social media authentically is better than using it in a more self-idealizing way, the overall effect of social media use on well-being might still be a negative. Future research could address this question by directly comparing no social media use to authentic social media use in both correlational and experimental settings.

Second, our findings do not provide any insights into why individuals might behave more or less authentically. For example, a deviation from the self-view might be explained by a lack of self-awareness, or an intentional misrepresentation of the self. It is possible that depending on whether deviation is driven by intent or not, authenticity might be more or less strongly related to well-being. That is, the psychological costs of deviating from one's self-view might be stronger when they are intentional such that the individual is fully aware of the fact that they are behaving in a self-idealizing way. Future research should explore this factor empirically.

Finally, the effects of authentic self-presentation on social media on well-being are robust but small $(\max(\beta) = 0.11)$ when compared to compared to other important predictors of well-being such as income, physical health, and marriage[47–49]. However, we argue that the effects described here are meaningful when trying to understand a complex and multifaceted construct such as Life Satisfaction. First, Study 1 captures authenticity using observations of actual behavior rather than self-reports. Given that such behavioral data captured in the wild do not suffer from the same response biases as self-reports which can inflate relationships between variables (e.g. common method bias[50]), and are often noisier than self-reports, their effect sizes cannot be directly compared[51]. In fact, the effect sizes obtained in Study 2 which was conducted in a much more controlled, experimental setting shows that the effect of authenticity on subjective well-being is substantially larger when measured with more traditional methods $(\max(d) = 0.45)$. In addition, while other factors such as employment and health are stronger predictors of well-being, they can be outside of the immediate control of the individual. In contrast, posting on social media in a way that is more aligned with an individual's personality is both up to the individual and relatively easy to change.

Social media is a pervasive part of modern social life[52]. Nearly 80% of Americans use some form of social media, and three quarters of users check these accounts on a daily basis[53]. Many have speculated that the artificiality of these platforms and their trend towards self-idealization can be detrimental for individual well-being. Our results suggest that whether or not engaging with social media helps or hurts an individual's well-being might be partly driven by how they use those platforms to express themselves. While it may be tempting to craft a self-enhanced Facebook presence, authentic self-expression on social media can be psychologically beneficial.

## Methods

**Study 1. Participants and procedure**. Data were collected through the MyPersonality project, an application available on Facebook between 2007 and 2012[31]. Users of the app completed validated psychometric tests including a measure of the Big Five personality traits[22,54], and received immediate feedback on their responses. A subsample of myPersonality users also agreed to donate their Facebook profile information—including their public profiles, their Facebook likes, their status updates, etc.—for research purposes. In addition, users could invite their Facebook friends to complete the personality questionnaire on their behalf, judging not their own personality but that of their friend.

To calculate authenticity, we developed a measure we refer to as Quantified Authenticity (QA). To compute this measure, we compared a person's self-reported personality to two external criteria: (1) their personality as predicted from Facebook Likes, and (2) their personality as predicted from the language used in their status updates (see "Measures" section below for more information). The number of participants varied between the two samples based on exclusionary criteria. To be included in the Language-based model, individuals had to have posted at least 500 words of Facebook status updates $(N = 3215)$. In the Likes-based model, only participants with 20 or more Likes were included $(N = 9237)$.

**Big Five personality**. Participants' personality was measured using the well-established Five Factor model of personality, also known as Big Five traits[54,55]. The Five Factor model posits five relatively stable, continuous personality traits: Openness to Experience, Conscientiousness, Extraversion, Agreeableness, and Neuroticism. The Big Five personality traits have been stable across cultures, instruments, and observers[56]. Additionally, years of research have linked them to a broad variety of behaviors, preferences and other consequential outcomes, including well-being[57] and behavior on Facebook[58].

**Self-reported personality**. Participants' views of their own personalities are based on the well-established International Personality Item Pool or IPIP[38]. Participants included in the analyses responded to 20–100 questions using a 5 point Likert-scale where 1 = strongly disagree to 5 = strongly agree.

**Computer-based predictions of personality from likes and status updates**. Recent methodological advances in machine learning have provided researchers with the ability to predict the personality of individuals from their social media

profiles[33–35]. Here, we used personality prediction of personality from Facebook Likes and the language used in status updates. For Facebook Likes ($N = 9327$), we obtained the personality predictions made by Youyou and colleagues[33], who used a 10-fold cross-validated LASSO regression to predict Big Five personality traits out of sample. On average, the predictions captured personality with an accuracy of $r = 0.56$ (correlation between predicted and self-reported scores). For status updates ($N = 3215$), we obtained the predictions made by Park et al.[34], who used cross-validated Ridge regression to infer personality from language features, such as individual words, combinations of words (n-grams), and topics. On average, the predictions captured personality with an accuracy of $r = 0.41$ (correlation between predicted and self-reported scores).

**Personality extremeness.** We calculated extremeness of participants' personality profiles as a control variable for our analyses by summing the absolute $z$-scores on all five traits. We include extremeness because extreme individual scores tend to produce larger absolute difference scores. Additionally, previous work has found that people with more extreme personality profiles might find it more difficult to blend into society and therefore experience lower levels of well-being[36].

**Self-ratings of well-being.** Individuals reported their Life Satisfaction—a key component of subjective well-being—on a five-item scale[39]. The SWLS has been shown to be a meaningful psychological construct, correlated with a number of important life outcomes such as marital status and health[59].

**Quantified Authenticity.** Quantified Authenticity was calculated in three steps. First, we $z$-standardized the personality scores on each of the three measures (self, Likes, language) to obtain a person's relative standing on the five personality traits in comparison to the reference group. Second, we computed the distance between self-reported personality and each of the externally inferred personality profiles using Euclidean distance, a widely established distance measure, which has been used in previous psychological research[36]. To make our measure more intuitively interpretable, we finally subtracted the distance measure from zero to obtain a measure of Quantified Authenticity for which higher scores indicate higher levels of authenticity. See Eq. (1) below.

$$\text{Quantified Authenticity} (x, y) = 0 - \sqrt{\sum_{i}^{5} (x_i - y_i)^2} \qquad (1)$$

For individual $i$, $x_i$ is the Cartesian coordinate of the self-view in a 5-dimensional personality space. For individual $i$, $y_i$ is the Cartesian coordinate of the language-, or likes-based personality. Our measure of Quantified Authenticity exhibited desirable level of variance, ranging all the way from highly authentic self-expression to considerable levels of self-idealization (see ridgeline plot of standardized Quantified Authenticity calculated based on Language and Likes in Supplementary Fig. 3). Additional information on the calculation of the three other metrics of Quantified Authenticity (i.e., Manhattan distance, correlational similarity, and cosine similarity) can be found in the SI.

**Study 2. Participants and procedure.** All study procedures were approved by the Columbia University Human Research Protection Office and informed consent was received from all study participants. Prior to completing the study, participants completed a pre-screening survey. This included a number of questions related to their social media activity and the BFI-2S as a measure of their Big Five personality traits[60]. Participants who qualified for the study were randomly assigned to one of two groups depicted as "Group A" and "Group B" in Fig. 3). Both groups received both interventions (authentic and self-idealized), however they received the treatments in a different order.

The study took place over the course of 2 weeks. On the first day of the study, participants received an email, which included the results of their personality test taken in the pre-screen. They then self-reported their baseline subjective well-being (t0). At the end of the survey, half of the students were asked to use the personality feedback to list three ways in which they could express themselves more authentically over the next week on social media. The second group was asked to list with three ways to express themselves in a more self-idealized way.

At the end of the first week, participants received an email with the second survey link. They completed the same subjective well-being measures (t1; Day 0–7), and were shown their personality feedback again as a reminder. The students who were previously assigned to the authentic condition were now asked to list three ways to express themselves in a more self-idealized way (based on their personality profile), and vice versa (reversing the intervention assignments). At the end of the second week, participants received an email with the final survey link. They completed the same subjective well-being measures (t2; Day 7–14).

**Subjective well-being.** Individuals reported their Life Satisfaction on the same five-item scale as Study 1[39]. In addition, participants responded to positive and negative affect[40] and a single-item general mood measure.

**Preregistration note.** We had pre-registered the use of the Positive and Negative Affect Scale[61]. However, due to an oversight of the research team, we accidentally collected data using the Brief Mood Inventory Scale[40]. In the SI, we replicate the results using a subset of items, which overlap between the BMIS and the PANAS-X. Given that the two scales are highly correlated, share the same format, and even share some of the same descriptors, we do not expect that the results would have been different when using the PANAS scale.

**Reporting summary.** Further information on research design is available in the Nature Research Reporting Summary linked to this article.

## Data availability

Data for Study 1 are available upon request to the authors. Data for Study 2 relevant to the analyses described are available on our OSF page (https://osf.io/fxav6/). Source data are provided with this paper.

## Code availability

Code to reproduce the analyses for Study 1 and Study 2 described herein is available on OSF (https://osf.io/fxav6/).

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

## Acknowledgements
We thank Blaine Horton, Jon Jachimowicz, Maya Rossignac-Milon, and Kostadin Kushlev for critical feedback which substantially improved this paper.

## Author contributions
E.R.B. and S.C.M. developed and designed the research; S.C.M. analyzed the data; E.R.B., S.C.M., W.Y., and S.I. interpreted the data and wrote the paper.

## Competing interests
The authors declare no competing interests.
