## [Peer Review File · Nature Communications]

Reviewers' Comments:

Reviewer #1:

Remarks to the Author:

The paper presents a correlation experiment on how users self-expression on Facebook is similar to their self-reported personality traits and how this resulting 'authenticity' score is correlated with self-reported life satisfaction. The self-expression on Facebook is quantified through predictive methods using Facebook likes or posted statuses, or as perceived by Facebook friends.

The correlation between life satisfaction and the resulting authenticity scores measured using the three methods is statistically significant, although the effect size is relatively small: .10 - .12 with no controls, .04-.06 with personality controls. If the paper argues that this effect size is important, then citations and comparisons to results from previous related studies should be mentioned.

The paper has only a single study and this may limit the generalizability of the findings. There is also a lack of stated hypotheses about the relationships that are studied and found.

The paper rests on several assumptions, which should all be stated (and most of them are), but these may have impact on the results.

One of these is that self-expression/self-view is just measured through personality. I would also have been curious to measure directly life satisfaction as perceived through text/likes/friends judging social media and compare this to self-reported life satisfaction.

Other comments:

- rather than combining all personality traits into a single score, separate correlations can be computed against difference for each personality trait. This may give extra insight into personality-specific relationships. This way, one could also measure any other interactions between each personality trait and life satisfaction (e.g. maybe appearing more extraverted is not related to life satisfaction, but appearing more introverted is related to life satisfaction)
- the Euclidian distance used ignores is there is higher variation in scores in one personality dimension compared to others and treats all dimensions as equally important. The other proposed metrics (cosine, Manhattan, correlation) are also similar in this aspect.
- I would like to see the correlation between the three measures of self-expression (friends, likes, text) in order to judge the robustness of the three correlations with QA
- the user samples are different from the three methods. Are these subsets representative of the general sample in terms of basic demographics and personality traits? How would the results look if just the intersection of the three sets of users is used?
- basic demographics (age/gender) should be controlled for in the experiments in case there are any age or gender effects
- previous research on the same data set showed that some closer friends are better than others at predicting personality. This may impact the overall metric. How many close friends are used? How predictive overall is this perceived personality of actual personality across the sample? If this is low, then the self-expression metrics would be off because people are objectively bad at personality perception rather than being misled by Facebook self-presentation

- how are friends judging the personality traits? Is this purely based on their social media profile or based (also) on real-life interactions? If the latter, then authenticity on social media in particular is not measured properly.
- for likes/status updates models - these were originally trained on myPersonality. Were these models retrained in order to make sure these users were not in the training data? Otherwise, the models may be overfitted to these particular users and the predictive performance would not be representative of new users.
- the interaction experiment lacks any hypotheses.
- what statistical test was used to quantify the normality of the QA variables?
- the last sentence of the results (217-218) hints at causality and needs to be rephrased.

Reviewer #2:

Remarks to the Author:

The authors presented a very crisp and clear study of personality expression in social media. The authors introduce a novel metric of Quantified Authenticity (QA), which is the difference between self-reported personality and other-reports, and find that higher levels of authenticity are related to small differences in life satisfaction.

In addition to being well-written and interesting, this study had several methodological strengths. Besides the large sample sizes, the authors used three different types of other-reports (by other humans or computer models), showing that the results were not dependent on any particular type of other-report. The QA metric was calculated in several different ways, as well, clearly showing that the results were not dependent on specific underlying distance metric.

The use of QA, the construct and measure at the heart of this study, raises several conceptual concerns. Currently, QA assumes that self-reported personality reflects the "authentic" self, while other-reports (by other humans or computer models via likes or language) reflect self-expression. Any discrepancies between these two estimates is interpreted as deviation from authenticity.

However, these interpretations of self- and other-reports need more justification, as they ignore equally plausible and existing interpretations from the personality literature. Consider self-reports - an alternative view is that self-reports reflect self-perceptions and self-judgments, which are generally useful and accurate, but also have small and systematic errors (Vazire & Carlson, 2011). If these systematic errors are not shared by other-reports, then the QA metric could be interpreted as a measure of someone's ability to judge themselves accurately, i.e., of self-knowledge vs. lack of self-awareness, rather than authenticity.

Other-reports may also differ from self-reports for reasons besides inauthenticity. Aside from random prediction error, other-predictions may differ systematically from self-reports in some cases because some people are simply harder to judge, and other researchers have suggested that the "judgability" of individuals may be related to psychological well-being (e.g., Colvin, 1993; see Funder, 1995, p. 661 on what makes a "good target"). With this in mind, the QA metric could be interpreted as a measure of the judgability of a target, rather than authenticity.

These alternate interpretations do not rule out that of the authors', but hopefully motivate some additional justification or validation behind the interpretation of the QA metric as a measure of authenticity. From a reader's perspective, I know very little about this metric before it is included into regression models. For example, if QA is a measure of authenticity, should it have predictable convergent or divergent correlations with other measures? Additional validation the QA construct is needed to support the authors' interpretation of the results and the following discussion and conclusions.

I thank the authors for the opportunity to review this work, and for their consideration of the above comments.

Greg Park

References

Colvin, C. R. (1993). "Judgable" people: Personality, behavior, and competing explanations. *Journal of Personality and Social Psychology*, 64(5), 861-873.

Funder, D. C. (1995). On the accuracy of personality judgment: a realistic approach. *Psychological review*, 102(4), 652.

Vazire, S., & Carlson, E. N. (2011). Others sometimes know us better than we know ourselves. *Current Directions in Psychological Science*, 20(2), 104-108.

Reviewer #3:

Remarks to the Author:

This paper examines the association of life satisfaction with the discrepancy between someone's self-reported personality traits and their traits as estimated from Facebook content. The paper's key strengths include its interesting topic, large and diverse sample, and use of online behavior to estimate personality traits. However, I also have several concerns about the paper, and I'm not sure whether they could all be adequately addressed in a revision.

1. My most important concern is direction of causality. Throughout, the paper suggests that the analyses are testing the effects of social media authenticity on life satisfaction. However, the authors acknowledge that "given that our analyses are purely correlational, we are unable to make causal claims on the link between self-expression and well-being. It is possible, for example, that individuals who experience higher levels of well-being are more likely to express themselves authentically on social media" (p. 9). To be honest, I find this alternative explanation—that people satisfied with their lives feel comfortable expressing themselves authentically on social media, whereas dissatisfied people are more tempted to present an idealized version of themselves on social media—to be more plausible than the authors' interpretation. Experimental or longitudinal data would be needed to distinguish between these accounts, but absent such data I recommend that the authors consider both possible causal directions throughout the paper, not just in the Discussion.

2. The obtained associations between social media authenticity and life satisfaction are small ($B = .08-.11$) before controlling for the effects of individual traits, and *very* small (.03-.05) after controlling for trait effects (Table 1). The effects are even smaller for some alternative measures of authenticity

(Figure 1). However, the paper does not currently discuss the size of these effects. To address this issue, I recommend prominently noting that the effects of social media authenticity on life satisfaction, or vice versa, appear to be very small.

3. The paper analyzes three indicators of social media authenticity: peer-reports made using the myPersonality Facebook app, personality predictions made from Facebook likes, and personality predictions made from language used in status updates. The latter two indicators strike me as better indicators of *social media* authenticity than do the discrepancies between self-reports and peer-reports. As the paper notes, most Facebook friends also know each other offline (p. 3), and I would expect personality peer-reports, even those collected through a Facebook app, to reflect the target's offline behavior at least as much as their online behavior. (This may be less true today than it was at the time these data were collected, about 10 years ago.)

4. The paper operationalizes idealization as the total discrepancy between an individual's self-reported and Facebook-predicted personality traits, and argues that discrepancies in either direction (i.e., self-reporting higher or lower trait levels) should be interpreted as self-idealization. To be frank, I'm not convinced by this argument. Experts and non-experts agree that it is generally preferable to be extraverted, agreeable, conscientious, emotionally stable, and open to experience than to be introverted, disagreeable, unconscientious, neurotic, and close-minded. Moreover, research on volitional personality change shows that most people want to change in the socially desirable direction on each Big Five trait, and almost no one wants to change in a socially undesirable direction (e.g., Hudson & Roberts, 2014; Hudson & Fraley, 2015). To address this issue, I recommend referring to "discrepancy" rather than "idealization" throughout the paper, and/or repeating the analyses while taking the socially desirable vs. undesirable direction of discrepancies into account. (These analyses could be briefly summarized in the main text and fully reported in the SOM.)

5. The paper refers to "subjective well-being" and "well-being," but life satisfaction is only one key component of subjective well-being, alongside positive affect and negative affect. I therefore recommend referring to "life satisfaction" rather than "well-being" throughout the paper, and calling for future research to examine affective components of subjective well-being.

6. The paper includes personality extremeness as a control variable in all analyses, but does not discuss this variable until the very end of the paper (p. 12). I recommend discussing this variable earlier in the paper, and also repeating the analyses without including this control variable to test whether/how it affects the key authenticity-satisfaction associations. (These analyses could be briefly summarized in the main text and fully reported in the SOM.)

In sum, I think this paper has promise due to its interesting topic and its large and rich data set. However, I also think it could be substantially improved by further checks on the robustness of the results, as well as greater caution and nuance when interpreting these results.

Review signed by Christopher J. Soto

Reviewers' comments:

Reviewer #1 (Remarks to the Author):

The paper presents a correlation experiment on how users self-expression on Facebook is similar to their self-reported personality traits and how this resulting 'authenticity' score is correlated with self-reported life satisfaction. The self-expression on Facebook is quantified through predictive methods using Facebook likes or posted statuses, or as perceived by Facebook friends.

The correlation between life satisfaction and the resulting authenticity scores measured using the three methods is statistically significant, although the effect size is relatively small: .10 - .12 with no controls, .04-.06 with personality controls. If the paper argues that this effect size is important, then citations and comparisons to results from previous related studies should be mentioned.

Response #1

We thank the reviewer for this comment, and agree that we can make this more explicit in our discussion of effect sizes. We have added to the general discussion (pg. 14), in which we compare the effect sizes to other predictors of well-being and highlight the importance of considering these findings in light of the fact that Study 1 utilizes actual behavioral outcomes (rather than self-reported survey responses) as predictors. In addition, Study 2, which was conducted in a much more controlled experimental setting, suggests that the within-person effects are small to medium. This further supports our argument that the effects in Study 1 are partly smaller because they rely on real-world data that is noisy and less prone to common method bias. The text of this section is included below:

“Finally, the effects of authentic self-presentation on social media on well-being are robust but small ($\max(\beta) = .11$) when compared to other important predictors of well-being such as income, physical health, and marriage.⁴⁵⁻⁴⁷ However, we argue that the effects described here are meaningful when trying to understand a complex and multifaceted construct such as Life Satisfaction. First, Study 1 captures authenticity using observations of actual behavior rather than self-reports. Given that such behavioral data captured in “the wild” do not suffer from the same response biases as self-reports which can inflate relationships between variables (e.g. common method bias⁴⁸), and are often “noisier” than self-reports, their effect sizes cannot be directly compared⁴⁹. In fact, the effect sizes obtained in Study 2 which was conducted in a much more controlled, experimental setting shows that the effect of authenticity on subjective well-being is substantially larger when measured with more traditional methods ($\max(d)=.45$). In addition, while other factors such as employment and health are stronger predictors of well-being, they can be outside of the immediate control of the individual. However, posting on social media in a way that is more aligned with an individual’s personality is both up to the individual and relatively easy to change.”

The paper has only a single study and this may limit the generalizability of the findings. There is also a lack of stated hypotheses about the relationships that are studied and found.

Response #2

We have taken this concern very seriously and added a new experimental study to the paper which provides additional evidence for the effect. In addition, we have added our explicit hypothesis for Study 1, stating that, “we test the extent to which authentic self-expression of personality characteristics are related to Life Satisfaction, hypothesizing that greater authentic self-expression will be positively correlated with Life Satisfaction (pg. 4). Similarly, for Study 2, we describe the hypothesis as follows, “[s]pecifically, we hypothesized that individuals who post more authentically over the course of a week would self-report greater subjective well-being at the end of that week, both at the between and within-person level” (pg. 9).

The paper rests on several assumptions, which should all be stated (and most of them are), but these may have impact on the results.

One of these is that self-expression/self-view is just measured through personality. I would also have been curious to measure directly life satisfaction as perceived through text/likes/friends judging social media and compare this to self-reported life satisfaction.

Response #3

We agree with the reviewer that this would be an interesting avenue for future research, but believe that it is beyond the scope of the paper, and focuses on a somewhat different angle of authenticity. While the relationship between self-reported Life Satisfaction and predicted Life Satisfaction is interesting in its own right, we would like to keep the focus on the relationship between personality and personality expression which we believe more closely aligns with the definition of authenticity as the unobstructed operationalization of the core self (Kernis & Goldman, 2006).

Other comments

- rather than combining all personality traits into a single score, separate correlations can be computed against difference for each personality trait. This may give extra insight into personality-specific relationships. This way, one could also measure any other interactions between each personality trait and life satisfaction (e.g. maybe appearing more extraverted is not related to life satisfaction, but appearing more introverted is related to life satisfaction)

Response #4

We have added additional analyses for individual traits to Supplementary Materials (Figures S1 and 2 in the SI). Specifically, we use response surface analyses to visualize the interactions between individual self-reported and computer-predicted traits on Life Satisfaction.

However, in line with prior conceptualizations of psychological fit, we have retained the focus on the more holistic measure in the main manuscript. While focusing on individual traits might provide insights into the relative importance of traits, it has the disadvantage of ignoring important information about the “full picture” and the interplay of different traits. It is possible, for example, that a person shows a high authenticity on Extraversion but at the same time a very low authenticity on Neuroticism. By modelling the traits individually, these

nuances might mask effects of authentic expression on Life Satisfaction.

- the Euclidian distance used ignores is there is higher variation in scores in one personality dimension compared to others and treats all dimensions as equally important. The other proposed metrics (cosine, Manhattan, correlation) are also similar in this aspect.

Response #5

The reviewer is correct in pointing out that by z-standardizing the personality scores we eliminate information about the natural variance of personality scores. However, we do not believe that this is necessarily a disadvantage as it is unclear whether the relative or absolute distance on a trait should receive more weight in the calculation of QA. In fact, giving more weight to traits of high variance would not necessarily lead to a more accurate estimate of authenticity, as small deviations from a trait that is more uniform in the population might be felt just as strongly by an individual. For example, if there is a wide range in Extraversion, deviating from my true self might not be as detrimental. However, if there is a small range and everybody is relatively uniform, I might feel the downside of acting out of character more strongly.

- I would like to see the correlation between the three measures of self-expression (friends, likes, text) in order to judge the robustness of the three correlations with QA

Response #6

In order to address this question, we first limited the sample to participants who are at the intersection of the Likes and Language-based models ($N = 1,711$). In this subset, we are able to compare the predicted personality traits in both models, comparing Likes-based Extraversion to Language-based Extraversion, and so on. We find that all five of the predicted traits from the Likes-based model are positively and significantly correlated with the same trait from the Language-based model at the $p < .001$ level (Openness: $r = .33$; Conscientiousness: $r = .42$; Extraversion: $r = .39$; Agreeableness: $r = .36$; Neuroticism: $r = .37$). Additionally, both measures of QA (Likes-based and Language-based) are positively and significantly correlated with one another ($r = .29, p < .001$). We have added the full correlation table of these results to the SI (Table S6, pg. 15). The fact that correlations are substantial, but not extremely high indicates that each of the measures might capture unique variance in a person's self-expression. Importantly, the finding that authenticity is predictive across both measures thus strengthens the robustness and generalizability of our findings.

- the user samples are different from the three methods. Are these subsets representative of the general sample in terms of basic demographics and personality traits? How would the results look if just the intersection of the three sets of users is used?

Response #7

We were able to obtain age and gender for a subset of our participants for Study 1 (Likes-Based, $N = 6,648$; Language-Based, $N = 2,943$). In this subset, the Likes-based and Language-based samples were similar in terms of gender (Likes-based: 61.24% female; Language-based:

58.27% female) and age (Likes-based: mean age = 24.97 years, SD age = 8.42 years; Language-based: mean age = 25.97 years, SD age = 10.42 years).

Additionally, as suggested by the reviewer, we replicated the results from the main text of Study 1 using this common subset of participants from the Likes-based and Language-based models described in Response #6 ($N = 1,711$). We find that QA is a positive and significant predictor of Life Satisfaction in this subset of participants (see Table S5, pg. 10).

- basic demographics (age/gender) should be controlled for in the experiments in case there are any age or gender effects

Response #8

As described in Response #7, we were able to obtain age and gender for a subset of our participants (Likes-Based, $N = 6,648$; Language-Based, $N = 2,943$). Given that this reduced the sample size considerably, we report the replication of our finding that QA predicts Life Satisfaction in the Supplementary Material (Table S3 on pg. 7). As we describe in the SI, “[w]e found that in the Likes-based model the effect of QA predicting Life Satisfaction was robust to the inclusion of gender and age controls in both Models 1 and 2. In the Language-based model, the effect of QA on Life Satisfaction remained significant in Model 1 but became non-significant in Model 2.”

- previous research on the same data set showed that some closer friends are better than others at predicting personality. This may impact the overall metric. How many close friends are used? How predictive overall is this perceived personality of actual personality across the sample? If this is low, then the self-expression metrics would be off because people are objectively bad at personality perception rather than being misled by Facebook self-presentation

- how are friends judging the personality traits? Is this purely based on their social media profile or based (also) on real-life interactions? If the latter, then authenticity on social media in particular is not measured properly.

Response #9

We thank the reviewer for this critical reflection on the comparison between self-rated personality and other, human-rated personality that we had included in the original version of the manuscript. Based on the comment, we have removed the Other-based model and analyses from the manuscript and instead focus on the algorithmically predicted personality from Likes and status updates on Facebook which capture self-expression on social media more directly.

- for likes/status updates models - these were originally trained on myPersonality. Were these models retrained in order to make sure these users were not in the training data? Otherwise, the models may be overfitted to these particular users and the predictive performance would not be representative of new users.

Response #10

Thank you for the chance to clarify this point. These were cross-validated predictions such that predicted users were not in the training sample. We have added this point to the description of the metrics (pg. 15). This section is included below:

“For Facebook Likes ($N=9,327$), we obtained the personality predictions made by Youyou and colleagues³², who used a 10-fold cross validated LASSO regression to predict Big Five personality traits out of sample. On average, the predictions captured personality with an accuracy of $r=.56$ (correlation between predicted and self-reported scores). For status updates ($N=3,054$), we obtained the predictions made by Park et al.³³, who used cross-validated Ridge regression to infer personality from language features, such as individual words, combinations of words (n-grams) and topics.”

- the interaction experiment lacks any hypotheses.

Response #11

As we state in the manuscript, the interaction analyses were exploratory and therefore did not have strong hypotheses. However, as we discuss in the introduction the rationale for this analysis was that people with socially desirable profiles (high Openness, Extraversion, Conscientiousness and Agreeableness, low Neuroticism) might benefit from authentic self-expression more than others. We have added this exploratory language more explicitly in the main text, stating in the Introduction that “[i]n exploratory analyses, we also explore whether this relationship is moderated by the personality characteristics of the individual” (pg. 4). Additionally, we added the following to the Results section of Study 1, “[i]n exploratory analyses, we considered whether authenticity might benefit individuals of different personalities differentially” (pg. 8).

- what statistical test was used to quantify the normality of the QA variables?

Response #12

Thank you for this point of clarification. We had previously referred to the QA measure as being relatively normal based on the Ridgeline plots (Figure S3 in the SI, pg. 16). However, upon conducting a Shapiro-Wilk normality test, it was revealed that that they were not normally distributed in either model (Likes-based or Language-based), and therefore we have updated that point from the manuscript, stating now that “[o]ur measure of Quantified Authenticity exhibited desirable level of variance, ranging all the way from highly authentic self-expression to considerable levels of self-idealization” (pg. 17).

- the last sentence of the results (217-218) hints at causality and needs to be rephrased.

Response #13

In the revision, we have embarked on Study 2 to speak to the issue of causality. In addition, we reviewed the statements in the manuscript to ensure that those referring to the correlational evidence in Study 1 reflect correlations. We also added the following sentence to the Results section of Study 1 stating that, “[t]he findings of Study 1 provide evidence for the link between authenticity on social media and well-being in a setting of high external validity. However, given the correlational nature of the study, we cannot make any claims about the causality of the effects” (pg. 8). Additionally, we add that the causal link between authentic self-expression and subjective well-being motivated the need for Study 2, stating “[t]o provide evidence for the directionality of our effect, we follow up the findings of Study 1 with a longitudinal experiment in Study 2” (pg. 8).

Reviewer #2 (Remarks to the Author):

The authors presented a very crisp and clear study of personality expression in social media. The authors introduce an novel metric of Quantified Authenticity (QA), which is the difference between self-reported personality and other-reports, and find that higher levels of authenticity are related to small differences in life satisfaction.

In addition to being well-written and interesting, this study had several methodological strengths. Besides the large sample sizes, the authors used three different types of other-reports (by other humans or computer models), showing that the results were not dependent on any particular type of other-report. The QA metric was calculated in several different ways, as well, clearly showing that the results were not dependent on specific underlying distance metric.

Response #1

We thank the reviewer for this positive evaluation of the paper which increased our motivation for the suggested revisions.

The use of QA, the construct and measure at the heart of this study, raises several conceptual concerns. Currently, QA assumes that self-reported personality reflects the “authentic” self, while other-reports (by other humans or computer models via likes or language) reflect self-expression. Any discrepancies between these two estimates is interpreted as deviation from authenticity.

However, these interpretations of self- and other-reports need more justification, as they ignore equally plausible and existing interpretations from the personality literature. Consider self-reports - an alternative view is that self-reports reflect self-perceptions and self-judgments, which are generally useful and accurate, but also have small and systematic errors (Vazire & Carlson, 2011). If these systematic errors are not shared by other-reports, then the QA metric could be interpreted as a measure of someone’s ability to judge themselves accuracy, i.e., of self-knowledge vs. lack of self-awareness, rather than authenticity.

Response #2

Thank you for this insightful feedback. We agree with your assessment that the exact nature of authenticity cannot be fully understood in the context of Study 1. We have acknowledged this shortcoming in the limitation section of the discussion (p. 14) which reads “Second, our findings do not provide any insights into why individuals might behave more or less authentically. For example, a deviation from the self-view might be explained by a lack of self-awareness, or an intentional misrepresentation of the self. It is possible that depending on whether deviation is driven by intent or not, authenticity might be more or less strongly related to well-being. That is, the psychological costs of deviating from one’s self-view might be stronger when they are intentional such that the individual is fully aware of the fact that they are behaving in a self-idealizing way. Future research should explore this factor empirically.”

We believe that the concern is partially alleviated by the fact that we have now excluded the

other-ratings in the new submission. This circumvents the problem of other people having potentially more reliable estimates of one's personality as the computer-based predictions are more directly grounded in the cues explicitly expressed and communicated by participants on social media.

Most importantly, our experiment does not suffer from the same limitation as participants are explicitly told to be authentic (vs. self-idealized in their posting). Given that we observe the similar effects on subjective well-being, we are more confident that QA is operating similarly to self-reported authenticity.

Other-reports may also differ from self-reports for reasons besides inauthenticity. Aside from random prediction error, other-predictions may differ systematically from self-reports in some cases because some people are simply harder to judge, and other researchers have suggested that the "judgability" of individuals may be related to psychological well-being (e.g., Colvin, 1993; see Funder, 1995, p. 661 on what makes a "good target"). With this in mind, the QA metric could be interpreted as a measure of the judgability of a target, rather than authenticity.

Response #3

We thank the reviewer for this thoughtful comment. Based on this comment, and similar thoughts from the other reviewers about the limitations and potential biases in the other-ratings of personality, we have removed the other-rating model from the manuscript and instead focus on the Quantified Authenticity relative to the Likes- and Language-based personality predictions. On a more conceptual level, we would argue that judgability is, in part, a consequence of authentic self-expression, and therefore difficult to disentangle empirically.

These alternate interpretations do not rule out that of the authors', but hopefully motivate some additional justification or validation behind the interpretation of the QA metric as a measure of authenticity. From a reader's perspective, I know very little about this metric before it is included into regression models. For example, if QA is a measure of authenticity, should it have predictable convergent or divergent correlations with other measures? Additional validation the QA construct is needed to support the authors' interpretation of the results and the following discussion and conclusions.

Response #4

As mentioned earlier, in Study 2 we find that when authentic self-expression on social media is directly manipulated, that is by telling participants to post in a way that is "authentic", we see similar patterns of increased subjective well-being. This provides some evidence that the Quantified Authenticity measure functions similarly to a more traditional authentic metric.

We would expect that the Quantified Authenticity (QA) measure would have convergent and discriminant validity compared to other authenticity measures. Part of the motivation for this research is the observed weaknesses of existing subjective authenticity measures, in part due to their reliance on self-reported data. On page 4, we mention that past measures of authenticity

have been shown to be biased by valence states and social desirability, making comparisons between QA and these existing measures less useful. Our hope is that by taking a step back from self-reported subjective authenticity, we are able to more clearly identify the definition of authenticity as the operation of the core self. We expect that the QA measure would positively correlate with self-reported authenticity, as well as with related constructs such as sincerity and self-esteem, and hope that future authenticity researchers further investigate how QA relates to similar constructs.

I thank the authors for the opportunity to review this work, and for their consideration of the above comments.

Greg Park

References

Colvin, C. R. (1993). "Judgable" people: Personality, behavior, and competing explanations. *Journal of Personality and Social Psychology*, 64(5), 861-873.

Funder, D. C. (1995). On the accuracy of personality judgment: a realistic approach. *Psychological review*, 102(4), 652.

Vazire, S., & Carlson, E. N. (2011). Others sometimes know us better than we know ourselves. *Current Directions in Psychological Science*, 20(2), 104-108.

Reviewer #3 (Remarks to the Author):

This paper examines the association of life satisfaction with the discrepancy between someone's self-reported personality traits and their traits as estimated from Facebook content. The paper's key strengths include its interesting topic, large and diverse sample, and use of online behavior to estimate personality traits. However, I also have several concerns about the paper, and I'm not sure whether they could all be adequately addressed in a revision.

1. My most important concern is direction of causality. Throughout, the paper suggests that the analyses are testing the effects of social media authenticity on life satisfaction. However, the authors acknowledge that "given that our analyses are purely correlational, we are unable to make causal claims on the link between self-expression and well-being. It is possible, for example, that individuals who experience higher levels of well-being are more likely to express themselves authentically on social media" (p. 9). To be honest, I find this alternative explanation—that people satisfied with their lives feel comfortable expressing themselves authentically on social media, whereas dissatisfied people are more tempted to present an idealized version of themselves on social media—to be more plausible than the authors' interpretation. Experimental or longitudinal data would be needed to distinguish between these accounts, but absent such data I recommend that the authors consider both possible causal directions throughout the paper, not just in the Discussion.

Response #1

We thank the reviewer for the comment. While we had mentioned this shortcoming as a limitation in the discussion section, we understand the reviewer's desire for additional evidence. Based on this feedback, we conducted experimental, longitudinal study to provide causal evidence for the relationship between authentic (vs. self-idealized) posting behavior on social media. The study design and analyses were pre-registered. Participants were randomly assigned to an authentic or self-idealized treatment, asking them to post in an authentic or self-idealized way over the course of a week. We then observed well-being outcomes at the end of the week. The treatments were then reversed in the following week to expand the study to a within-subjects design. We found within-subject effects of self-idealization vs. authentic posting on social media such that in the weeks where participants posted authentically, they reported more positive mood, positive affect, and marginally significantly less negative affect. We believe this strengthens our causal argument from authentic self-expression on social media to subjective well-being.

2. The obtained associations between social media authenticity and life satisfaction are small ($B = .08-.11$) before controlling for the effects of individual traits, and *very* small (.03-.05) after controlling for trait effects (Table 1). The effects are even smaller for some alternative measures of authenticity (Figure 1). However, the paper does not currently discuss the size of these effects. To address this issue, I recommend prominently noting that the effects of social media authenticity on life satisfaction, or vice versa, appear to be very small.

Response #2

Thank you. This is a sentiment shared by Reviewer 1. We have now added an explicit discussion of the effect sizes observed in Study 1 and Study 2 to the Discussion section of the manuscript, which we have included below:

“Finally, the effects of authentic self-presentation on social media on well-being are robust but small ($\max(\beta) = .11$) when compared to other important predictors of well-being such as income, physical health, and marriage.⁴⁵⁻⁴⁷ However, we argue that the effects described here are meaningful when trying to understand a complex and multifaceted construct such as Life Satisfaction. First, Study 1 captures authenticity using observations of actual behavior rather than self-reports. Given that such behavioral data captured in “the wild” do not suffer from the same response biases as self-reports which can inflate relationships between variables (e.g. common method bias⁴⁸), and are often “noisier” than self-reports, their effect sizes cannot be directly compared⁴⁹. In fact, the effect sizes obtained in Study 2 which was conducted in a much more controlled, experimental setting shows that the effect of authenticity on subjective well-being is substantially larger when measured with more traditional methods ($\max(d)=.45$). In addition, while other factors such as employment and health are stronger predictors of well-being, they can be outside of the immediate control of the individual. However, posting on social media in a way that is more aligned with an individual’s personality is both up to the individual and relatively easy to change.”

3. The paper analyzes three indicators of social media authenticity: peer-reports made using the myPersonality Facebook app, personality predictions made from Facebook likes, and personality predictions made from language used in status updates. The latter two indicators strike me as better indicators of *social media* authenticity than do the discrepancies between self-reports and peer-reports. As the paper notes, most Facebook friends also know each other offline (p. 3), and I would expect personality peer-reports, even those collected through a Facebook app, to reflect the target’s offline behavior at least as much as their online behavior. (This may be less true today than it was at the time these data were collected, about 10 years ago.)

Response #3

This is an excellent point. While we find that the effects were relatively similar across the three measures, it is certainly likely that other-raters knew participants in the sample in an offline context as well. Based on this and similar feedback from the other reviewers regarding the other-rated personality model, we have removed it from the manuscript limit our claims to authentic self-expression on social media.

4. The paper operationalizes idealization as the total discrepancy between an individuals’ self-reported and Facebook-predicted personality traits, and argues that discrepancies in either direction (i.e., self-reporting higher or lower trait levels) should be interpreted as self-idealization. To be frank, I’m not convinced by this argument. Experts and non-experts agree that it is generally preferable to be extraverted, agreeable, conscientious, emotionally stable, and open to experience than to be introverted, disagreeable, unconscientious, neurotic, and close-minded. Moreover, research on volitional personality change shows that most people want to change in

the socially desirable direction on each Big Five trait, and almost no one wants to change in a socially undesirable direction (e.g., Hudson & Roberts, 2014; Hudson & Fraley, 2015). To address, this issue, I recommend referring to “discrepancy” rather than “idealization” throughout the paper, and/or repeating the analyses while taking the socially desirable vs. undesirable direction of discrepancies into account. (These analyses could be briefly summarized in the main text and fully reported in the SOM.)

Response #4

We thank the reviewer for his feedback on this methodological consideration. In fact, we had discussed this point in great length while setting up the manuscript. We agree that “on average” people have a notion of which traits are socially desirable, and which direction they might want to change in, and have acknowledged so more directly by adding the suggested citations to the manuscript. Specifically, we state that, “Although research suggests that there are certain personality traits that are more desirable on average (Hudson & Roberts, 2014; Hudson & Fraley, 2015), the extent to which a person sees scoring high or low on a given trait is likely somewhat idiosyncratic and depends – at least in part – on other people in their social network. For example, behaving in a more extraverted way might be self-enhancing for most people; however, there might be individuals for whom behaving in a more introverted way might be more desirable (e.g. because the norm of their social network is more introverted). Hence, our conceptualization of Quantified Authenticity allows for deviations in different direction” (pg. 5).

However, we still believe that prescribing the direction of desired change would ignore individual differences in the ways in people think of themselves as well as their ideal selves. While we believe that our measure of authenticity in the context of social media posting is reflective of self-idealization, we have toned down the language around self-idealization in the context of Study 1 and have instead highlighted the self-discrepancy aspect. We still retain the explanation for why we believe this captured self-idealization.

To alleviate the reviewer’s concerns, we have added two graphs to the supplementary information showing the links between self-idealization and Life Satisfaction in the normative way suggested by the reviewer (SI, Figure S4A-B, pg. 18). That is, positive values on self-idealization are defined as higher scores on Openness, Conscientiousness, Extraversion and Agreeableness, but lower scores in Neuroticism (the measure averages across the five traits). As can be seen from the different slopes fitted for self-deprecation individuals (red), and self-idealizing individuals (green), the effect of authenticity on Life Satisfaction is driven in particular by the negative effects of self-idealization.

5. The paper refers to “subjective well-being” and “well-being,” but life satisfaction is only one key component of subjective well-being, alongside positive affect and negative affect. I therefore recommend referring to “life satisfaction” rather than “well-being” throughout the paper, and calling for future research to examine affective components of subjective well-being.

Response #5

We agree with the reviewer that our use of the term well-being was rather liberal in the original manuscript in which we only studies Life Satisfaction. However, given that Study 2 now includes additional affective components of subjective well-being, we believe retaining the terminology around well-being is justified.

6. The paper includes personality extremeness as a control variable in all analyses, but does not discuss this variable until the very end of the paper (p. 12). I recommend discussing this variable earlier in the paper, and also repeating the analyses without including this control variable to test whether/how it affects the key authenticity-satisfaction associations. (These analyses could be briefly summarized in the main text and fully reported in the SOM.)

Response #6

We have added an explicit mention of the “personality extremeness” variable earlier in the main text (pg. 6) stating that, “Additionally, we included a control variable for an individual’s most extreme trait, given that people with more extreme personality profiles might find it more difficult to blend into society and therefore experience lower levels of well-being.³⁵”

The relationships between Quantified Authenticity and Life Satisfaction are statistically significant and positive in 6 out of 8 models with no controls (Model 1). This is referenced in Figure 1 and is also available in Table S4 in the SI (pg. 9).

In sum, I think this paper has promise due to its interesting topic and its large and rich data set. However, I also think it could be substantially improved by further checks on the robustness of the results, as well as greater caution and nuance when interpreting these results.

Review signed by Christopher J. Soto

Reviewers' Comments:

Reviewer #2:

Remarks to the Author:

With this revision, the authors have sufficiently addressed my comments on the first version of the manuscript. I appreciate the authors' thoughtful and detailed responses to my initial concerns.

In particular, the added experimental study considerably strengthens this paper by more directly tying a much clearer example of authenticity/idealism to multiple measures of subjective well-being. I also found the supplemental analysis of self-idealism vs self-deprecation very interesting (and surprising).

I have no additional concerns about this revision.

Greg Park

Reviewer #3:

Remarks to the Author:

This paper examines the effects of authentic vs. idealized social media self-expression on subjective well-being. The paper's key strengths include its interesting topic, large and diverse sample and use of online behavior to estimate personality traits in Study 1, and experimental design of Study 2.

I reviewed the original version of this manuscript as Reviewer 3, and raised several concerns. I was therefore pleased to see that the revised manuscript thoughtfully addresses all of my points. Perhaps most importantly, the addition of Study 2 substantially strengthens the claim that social media self-expression has a causal effect on subjective well-being. The revised manuscript also explicitly discusses effect size, reports supplemental analyses of authenticity that take the direction of discrepancy into account, broadens the measurement of well-being in Study 2, clarifies the effects of personality extremeness on the key results of Study 1, and removes self-peer agreement as an indicator of social media authenticity. All of these changes strengthen the paper.

My only remaining suggestion concerns the new analyses of normative self-enhancement (i.e., rating oneself as more extraverted, agreeable, conscientiousness, emotionally stable, and open-minded than is indicated by one's Facebook behavior) vs. self-deprecation (i.e., rating oneself as less extraverted, agreeable, etc.). As shown in Supplemental Figure 4 and discussed in Response #4 to Reviewer 3, these analyses indicate that normative self-enhancement has a negative effect on well-being, whereas normative self-deprecation has no effect. It therefore seems to be self-enhancement specifically, rather than overall self-discrepancy/lack of authenticity, that harms subjective well-being. This point seems important enough to discuss in the main text, rather than relegating it to the SI.

Overall, however, I commend the authors on a thoughtful revision.

Review signed by Christopher J. Soto

***REVIEWERS' COMMENTS:

Reviewer #2 (Remarks to the Author):

With this revision, the authors have sufficiently addressed my comments on the first version of the manuscript. I appreciate the authors' thoughtful and detailed responses to my initial concerns.

In particular, the added experimental study considerably strengthens this paper by more directly tying a much clearer example of authenticity/idealism to multiple measures of subjective well-being. I also found the supplemental analysis of self-idealism vs self-deprecation very interesting (and surprising).

I have no additional concerns about this revision.

Greg Park

Reviewer #3 (Remarks to the Author):

This paper examines the effects of authentic vs. idealized social media self-expression on subjective well-being. The paper's key strengths include its interesting topic, large and diverse sample and use of online behavior to estimate personality traits in Study 1, and experimental design of Study 2.

I reviewed the original version of this manuscript as Reviewer 3, and raised several concerns. I was therefore pleased to see that the revised manuscript thoughtfully addresses all of my points. Perhaps most importantly, the addition of Study 2 substantially strengthens the claim that social media self-expression has a causal effect on subjective well-being. The revised manuscript also explicitly discusses effect size, reports supplemental analyses of authenticity that take the direction of discrepancy into account, broadens the measurement of well-being in Study 2, clarifies the effects of personality extremeness on the key results of Study 1, and removes self-peer agreement as an indicator of social media authenticity. All of these changes strengthen the paper.

My only remaining suggestion concerns the new analyses of normative self-enhancement (i.e., rating oneself as more extraverted, agreeable, conscientiousness, emotionally stable, and open-minded than is indicated by one's Facebook behavior) vs. self-deprecation (i.e., rating oneself as less extraverted, agreeable, etc.). As shown in Supplemental Figure 4 and discussed in Response #4 to Reviewer 3, these analyses indicate that normative self-enhancement has a negative effect on well-being, whereas normative self-deprecation has no effect. It therefore seems to be self-enhancement specifically, rather than overall self-discrepancy/lack of authenticity, that harms subjective well-being. This point seems important enough to discuss in the main text, rather than relegating it to the SI.

We have now added the below section to the manuscript (pg. 6):
--

To further explore the mechanisms of Quantified Authenticity, we conducted analyses that distinguished between normative self-enhancement (i.e., rating oneself as more extraverted, agreeable, conscientiousness, emotionally stable, and open-minded than is indicated by one's Facebook behavior) from self-deprecation (i.e., rating oneself lower on all of these traits). The analyses indicate that normative self-enhancement has a negative effect on well-being, whereas normative self-deprecation has no effect. These findings suggest that self-enhancement specifically, rather than overall self-discrepancy/lack of authenticity, is detrimental to subjective well-being (see Supplementary Figure S4).

Overall, however, I commend the authors on a thoughtful revision.

Review signed by Christopher J. Soto